# COVID-19 Vaccinations, Infections, and Outcomes Among 784 People Living with HIV

**DOI:** 10.3390/v16121805

**Published:** 2024-11-21

**Authors:** Keren Mahlab-Guri, Irina Komarova, Laliv Kadar, Shay Nemet, Ramon Cohen, Sara Radian-Sade, Achiel Tova, Alex Guri, Shira Rosenberg-Bezalel, Daniel Elbirt

**Affiliations:** 1Faculty of Medicine, Hebrew University of Jerusalem, Jerusalem 9112102, Israel; lalivka@clalit.org.il (L.K.); shayne2@clalit.org.il (S.N.); alexgur@clalit.org.il (A.G.); shiraroz@clalit.org.il (S.R.-B.); danielel@clalit.org.il (D.E.); 2Allergy, Clinical Immunology and AIDS Center, Kaplan Medical Center, Rehovot 7680400, Israel; irinako@bmc.gov.il (I.K.); sara_rs@clalit.org.il (S.R.-S.); tovaac@clalit.org.il (A.T.)

**Keywords:** HIV, AIDS, COVID-19, COVID-19 mRNA vaccine, vaccination response

## Abstract

Introduction: Variants of COVID-19 are responsible for 700 million infections and 7 million deaths worldwide. Vaccinations have high efficiency in preventing infection and secondary benefits of reducing COVID-19 hospital admissions, attenuating disease severity and duration of illness. Conflicting reports were published regarding COVID-19 among PLWH. Objective: The aim of this study was to evaluate COVID-19 morbidity, hospitalization, and the magnitude of immunological response to sequential BNT 162b2 mRNA vaccines in PLWH regarding demographic and clinical factors. Results: Our retrospective study included 784 PLWH who had at least one anti- SARS-CoV-2 antibody test between March 2021 and October 2021. Half of our patients (392) had CD4 cell counts above 500 cells/µL, 40.2% (315) had 200 < CD4 < 500 cells/µL and only 9.8% (77) had CD4 < 200 cells/µL at their last laboratory workup. The mean age was 50.2 ± 12.2 years. About 90% of our patients were given at least two doses of the BNT 162b2 Pfizer vaccines; about 60% received three doses of the vaccine. About a quarter of our patients (27.6%) had COVID-19 infection. Only six patients required hospital admission. All six patients recovered from COVID-19 infection. Titers of COVID-19 antibodies were lower for patients with CD4 cell counts of less than 200 cells/µL in the first, second, and third serological tests with statistical significance. In a multinomial logistic regression, the influence of other factors such as age, sex, and previous COVID-19 infection on first COVID-19 antibody titers was not significant. Conclusions: PLWH are responsive to COVID-19 vaccines. As was expected, patients with higher CD4 cell counts had higher titers of COVID-19 antibodies and lower hospitalization rate. Age, sex, and previous COVID-19 infection did not significantly affect antibody titers according to our study. Larger prospective studies with control groups are needed to further characterize immunologic response to COVID-19 vaccination among PLWH.

## 1. Introduction

First cases of corona virus disease 2019 (COVID-19) emerged in Wuhan, China in December 2019. The disease, caused by the severe acute respiratory syndrome coronavirus 2 (SARS-CoV-2), spread rapidly worldwide and was announced as a global pandemic by the World Health Organization (WHO) in March 2020. As of July 2023, the different variants of COVID-19 have been responsible for 700 million infections and 7 million deaths. At the end of 2020, two mRNA COVID-19 vaccines were authorized by the FDA. Large, randomized, controlled phase 3 trials proved two doses of BNT 162b2 Pfizer vaccine to confer 95% protection against COVID-19 infection [1]. Vaccination in real world conditions reconfirmed the high efficiency in preventing infection and demonstrated secondary benefits of the vaccine in reducing COVID-19 hospital admissions, attenuating disease severity and duration of illness [2]. Our study was a real-life study from Israel that contributes to the growing body of evidence that suggests PLWH, particularly those with well-managed HIV, exhibit a robust immunologic response to COVID-19 vaccination.

Patients living with HIV (PLWH) were assumed to have an increased risk of COVID-19 infection and to have more severe illness at the beginning of COVID-19 pandemic. A report by the World Health Organization (WHO) found 15,522 PWH and 168,649 adults without known HIV infection were hospitalized with suspected or confirmed SARS-CoV-2. HIV was associated with an increased risk of severe or critical disease at admission compared to uninfected controls, after adjusting for age, sex, and underlying comorbidities. HIV was also independently associated with higher mortality risk [3]. In a review published in September 2021, an increased risk of severe COVID-19 disease progression, even in the setting of well-controlled HIV, was reported [4].

Contradicting reports were published later. A population-based study within New York State of 2988 PLWH who were diagnosed with SARS-CoV-2 demonstrated a similar incidence of infection to that of the general population after standardization (adjusted rate ratio 0.94 (95% CI, 0.91–0.97)) [5]. In a later study from the USA of 79,401 patients with COVID-19 and HIV and 1,564,875 with COVID-19 alone, patients with COVID-19 and HIV did not show a significant difference in mortality compared to those with COVID-19 alone (10.2% vs. 11.3%, respectively, *p* = 0.35). However, the rate of renal disease was higher among PLWH with COVID-19 infection compared to patients with COVID-19 infection alone [6].

PLWH are often less responsive to vaccines, which show lower efficacy in this group. Even successfully treated HIV patients show specific defects in memory follicular T helper cells that lead to reduced B cell response and antibody production [7,8,9].

Data regarding COVID-19 vaccination in PLWH, and the clinical characteristics of their serologic response, were sparse during the beginning of the COVID-19 pandemic [10,11,12,13]. In a study from China, 55 PLWH and 21 healthcare workers had comparable spike receptor binding domain IgG titers, but antibody responses were lower in poor immunological responders (CD4+ T cell counts < 350 cells/µL) compared with immunological responders (CD4^+^ T cell counts ≥ 350 cells/µL) [11]. In a study from Germany, a robust antibody response was observed in 665 PLWH undergoing standard vaccination against SARS-CoV-2. A higher CD4 cell count was associated with a higher concentration of antibodies [12].

The aim of our study was to evaluate the magnitude of immunological response to sequential BNT 162b2 mRNA vaccines in PLWH regarding demographic and clinical factors including CD4 status, viral load, COVID-19 morbidity, and hospitalization.

## 2. Methods

### 2.1. Patients

PLWH treated at the Neve Or HIV clinic were offered an anti- SARS-CoV-2 antibody test to be added to their routine laboratory workup. Testing for anti- SARS-CoV-2 antibodies started on 1 March 2021. Patients with a COVID-19 serology test between 1 March 2021 and 30 October 2021 were included in the current retrospective cohort.

### 2.2. Demographic, Clinical, and Laboratory Data

Data were collected from patients’ electronic files. Demographic characteristics included gender, age, and way of HIV acquisition. Clinical characteristic included CD4, nadir of CD4, time since HIV diagnosis (in years), viral suppression, COVID-19 morbidity, hospital admissions for COVID-19 infection and outcomes, and number and dates of COVID-19 vaccination and antibody titers. COVID-19 infection was defined as a positive result of RT-PCR testing via nasopharyngeal swab.

The BNT 162b2 Pfizer vaccine was used for all vaccinated patients and subsequent second and third booster immunizations, if taken. Abbott’s^©^ SARS-CoV-2 IgG II quant assay was used to determine the qualitative measure of IgG antibodies against spike receptor binding domain (RBD). This test detects a positive antibody response from both infected and vaccinated individuals. A titer of less than 50 reflects no immunity, titers between 50 and 40,000 are considered immune, and results higher than 40,000 are regarded as highly immune. RBD IgG levels were recorded in relation to the time (measured in days) from each vaccine taken.

Patients were divided to three groups according to their CD4 cell count: less than 200 cells/µL, 200–500 cells/µL, and above 500 cells/µL. Viral load was considered undetectable for participants with less than 200 copies/µL.

### 2.3. Statistical Analysis

Data are presented as means ± standard deviations. Continuous variables between the study groups were tested for normality by Shapiro–Wilk test and when abnormal distribution was found, non-parametric tests were performed. The Kruskal–Wallis H groups test was performed to compare the three groups. A repeated measures analysis of variance was used to determine any significant differences between variability over time. Initial multivariable logistic regression models were built, including variables with statistical significance at univariate analysis. A *p* value < 0.05 was considered statistically significant. Data were analyzed using SPSS 25.

## 3. Results

### 3.1. Demographic and Clinical Characteristics

Our retrospective study included 784 PLWH who had at least one anti- SARS-CoV-2 antibody test between the first of March 2021 and the end of October 2021. Demographic and clinical data for these patients are presented at Table 1. Half of our patients (392) had CD4 cell counts above 500 cells/µL, 40.2% (315) had 200 < CD4 < 500 cells/µL, and only 9.8% (77) had CD4 < 200 cells/µL at their last laboratory workup. The mean age of the patients was 50.2 ± 12.2 years; patients with CD4 cell counts of less than 200 cells/µL were older with statistical significance; *p* = 0.0001. Most of our patients (55.6%) were males; a higher number of males was in the group of patients with low CD4 cell counts, with statistical significance. Most of our HIV-positive patients (60.2%) were immigrants from countries where the illness is endemic. The mean duration of HIV infection was 16.5 ± 7.8 years. The CD4 nadir was lower for patients with lower CD4 cell counts, with statistical significance (Table 1); 93.9% of our patients had HIV viral suppression at their last laboratory workup and a lower rate of viral suppression was observed in the group of patients with lower CD4 cell counts.

### 3.2. COVID-19 Vaccination and Infection

As can be seen in Table 2, about 90% of our patients were given at least two doses of the BNT 162b2 Pfizer vaccines and about 60% received three doses of the vaccine. There was no significant difference between the groups of patients with CD4 cell counts below 200, between 200 and 500, and above 500 cells per microliter. About a quarter (27.6%) of our patients (217) had COVID-19 infection, with no correlation to their CD4 cell count.

First serological tests were taken after COVID-19 infection in 37.3% of those patients (81). Low titers (<50, no immunity) of antibodies against COVID-19 were found in 35/217 patients; only 3 of them were infected with COVID-19 before first titers were measured. High titers were found in 12/217 COVID-19 infected patients; only 2 of them were not infected before the first titer was measured.

About a quarter (27.4%) of our patients (215/784) had high COVID-19 antibody titers at the first measurement and only 10 of them (4.6%) were previously infected with COVID-19. About half (51.4%, or 403/784) had a normal antibody response to COVID-19 and only 68/403 (16.9%) of them were previously infected with COVID-19. Of the patients with normal or high titers, 12.6% (78/618) were previously infected with COVID-19.

Breakthrough COVID-19 infection (COVID-19 infection after three doses of vaccine) was detected in 70 patients: in one patient with CD4 < 200 (1/46, 2.1%), 33 patients with 200 < CD4 < 500 (33/189, 17.4%), and 36 patients with CD4 > 500 (36/235, 15.3%); *p* = 0.0001. Patients with breakthrough infection were younger compared to the entire cohort (47.6 ± 13 vs. 50.2 ± 12) but without statistical significance; *p* = 0.084.

Only six patients (0.76%) required hospital admission for their COVID-19 infection: one patient with a CD4 cell count of less than 200 cells/µL, five patients with a CD4 cell count of 200–500, and none with a CD4 cell count above 500 cells/µL; *p* = 0.047. All six patients recovered from COVID-19 infection. None of our patient died of COVID-19. Titers of COVID-19 antibodies were lower for patients with CD4 cell counts of less than 200 cells/µL in the first, second, and third serological tests with statistical significance; *p* = 0.0001 for the first titer (Table 2).

### 3.3. Antibody Titers over Time

As was expected, repeated measurements revealed an increase in the qualitative measure of IgG antibodies against spike receptor binding domain (RBD) over time, with statistical significance; *p* < 0.001 (Figure 1). The first measurement showed 78.8% (618/784) of our patients had a normal or high titer of COVID-19 antibodies, while 88.8% of our patients had a normal or high titer of COVID-19 antibodies at their second measurement (493/555). The third measurement revealed that 93.4% of our patients had normal or high titers of COVID-19 antibodies, but the group was small (43/46).

### 3.4. Multivariant Analysis

Multinomial logistic regression was performed to examine the influence of other factors such as age, sex, previous COVID-19 infection, and last CD4 value on COVID-19 antibody titers. The first titer of antibodies correlated to CD4 at the last laboratory workup. Low CD4 cell counts (<200 cells/µL) reduced the chances of normal and high titers of COVID-19 antibodies with an odds ratio of 0.27; *p* < 0.01 for normal antibody response and odds ratio of 0.304 for high antibody response; *p* = 0.001. Age, sex, and previous infection with COVID-19 did not have significant influence on first COVID-19 antibody titers. Multinomial logistic regression of the second titer of antibodies revealed that low CD4 cell counts (<200 cells/µL) were strongly associated with low COVID-19 antibody titers, odds ratio of 0.261; *p* = 0.002. Older age was associated with higher antibody titers with an odds ratio of 1.025 for every year; *p* = 0.04.

## 4. Discussion

In this study, which examined 784 people living with HIV (PLWH) in Israel, we observed a high rate of COVID-19 vaccination, with about 90% of the participants receiving at least two doses of the BNT 162b2 mRNA vaccine. The high vaccination rate against COVID-19 observed in our cohort is particularly noteworthy when compared to other studies in different regions. For instance, a cross-sectional study in South Africa reported that only 57% of PLWH were willing to accept future vaccination [14], and a study from New York City found that 28% of PLWH had not received any COVID-19 vaccination by March 2022 [15]. The rate of COVID-19 vaccination among our HIV patients was higher compared to the rate of vaccination among the general population in Israel, which was 84.3% in March 2022 [16]. The higher vaccination rates in our study may reflect the stronger engagement of PLWH to healthcare compared to the general population.

The demographic characteristics of our patients represents HIV-infected patients in a real world setting. Our cohort included a unique patient population with a relatively high proportion of women and individuals of African descent, mainly Ethiopian, which may differ from other studies that predominantly feature male participants [15]. Additionally, our study followed a large number of HIV patients during a long period of the COVID-19 pandemic, reflecting the surge of different variants waves including the beta, delta, omicron BA1/2, and even to some extent omicron BA4/5. This broad temporal coverage adds robustness to our findings, suggesting that the immune response observed was consistent across different variants of the virus.

In October 2021, the cumulative COVID-19 infection rate of our HIV patients was 27.6%, higher than the infection rate of the entire Israeli population at the same time point [17]. Despite the higher rate of infection, the hospitalization rate due to COVID-19 was not higher than the rate of hospitalization in the entire Israeli population [17]. Only six patients required hospitalization, all of them recovered. A USA study from the beginning of the COVID-19 pandemic reported that people living with diagnosed HIV experienced poorer COVID-related outcomes compared to persons living without diagnosed HIV [5]. A later study from the UK found an adjusted hazard ratio of 0.49 for invasive mechanical ventilation/death of hospitalized PLWH compared to the HIV-negative hospitalized population [18]. This encouraging finding suggests COVID-19 vaccination provides substantial protection against severe disease in PLWH, even among those with lower CD4 counts. The absence of COVID-19-related mortality in our cohort further supports the effectiveness of vaccination in this population.

In line with other studies, our research confirmed that PLWH with higher CD4 cell counts exhibited better immunologic responses to COVID-19 vaccines [19,20]. Specifically, patients with CD4 counts below 200 cells/µL had significantly lower antibody titers after vaccination [21,22]. Notably, our findings align with those from other studies, such as an Israeli study that reported 136 PLWH having a humoral immune response comparable with that of healthcare workers (without HIV) to two doses of BNT 162b2 mRNA vaccine. PLWH with CD4 cell counts less than 300 cells/µL had lower antibody titers than those with a higher CD4 cell count [16].

Breakthrough COVID-19 infection after three doses of BNT 162b2 mRNA vaccines were detected in 70 patients. The rate of breakthrough infection was significantly lower in the group of patients with a CD4 cell count < 200 (2.1%). This contrasts with a study by Xueying Yang et al. [23], who found a lower rate of breakthrough infection in patients with a high CD4 cell count. We can assume that the difference between the studies is related to the different time periods during the pandemic, with more cautious behavior, especially among the older and immunocompromised patients at an earlier stage.

Age and sex did not have significant impact on the first antibody titer in our cohort, as was shown in the multinomial logistic regression (Table 3). Surprisingly, in contradiction to previous studies such as the study by Mary Duro et al. [24], COVID-19 infection also did not significantly affect the first antibody titers against COVID-19 in our multinomial logistic regression. This can be explained by the fact that only 81 patients (10.3%) were infected with COVID-19 before the first titer was taken. The low rate of previously COVID-19-infected patients in our study masked the real effect of previous infection on antibody titers.

Older age was significantly associated with higher antibody titers in subsequent measurements. This finding contrasts with some reports that suggest older individuals might have a diminished response to vaccination [25]. The reasons for this discrepancy could be multifactorial and warrant further investigation. It is possible that the older participants in our study had more consistent healthcare access or that other unmeasured factors contributed to this outcome.

As is shown on Figure 1, repeated measurements in a small group of patients (46) revealed an increase in the qualitative measure of IgG antibodies against the spike receptor binding domain (RBD) over time, with statistical significance. The third measurement revealed that 93.4% of those patients had normal or high titers of COVID-19 antibodies. The increased antibody titers were probably the result of repeated vaccinations with BNT 162b2 mRNA vaccine, since only four patients in this group were diagnosed as positive for COVID-19. However, undiagnosed asymptomatic COVID-19 infection in parallel to the spread of the COVID-19 pandemic cannot be ruled out as the reason for the increase in the titers.

Our findings contribute to the growing body of evidence that suggests PLWH, particularly those with well-managed HIV, exhibit a robust immunologic response to COVID-19 vaccination [26]. An Israeli prospective study of 143 PLWH found the BNT 162b2 mRNA vaccine to be immunogenic in PLWH that were taking antiretroviral treatment, with unsuppressed CD4 cell counts and suppressed viral load [10]. A retrospective analysis of 665 PLWH from Germany/Munich found a strong immune response to standard vaccination with a high antibody concentration associated with being female and having high CD4 cell counts [12]. Our study found that a significant majority (78.8%) of participants had normal or high titers of COVID-19 antibodies after the first serological measurement, and this proportion increased with subsequent measurements.

Despite these positive outcomes, it is important to acknowledge the limitations of our study. As a retrospective analysis, our study is subject to the inherent biases associated with this study design, including selection bias and the potential for missing data. Moreover, our cohort was relatively unique, with a high rate of viral suppression and a specific demographic composition, which may limit the generalizability of our findings to other PLWH populations.

In conclusion, our study provides valuable insights into the immunologic response to COVID-19 vaccination in PLWH, highlighting the importance of vaccination in this population. The findings underscore the need for continued efforts to vaccinate PLWH, particularly those with lower CD4 counts, to ensure broad protection against COVID-19. Further research, particularly prospective studies with control groups, is needed to fully understand the long-term immunologic outcomes and the potential need for additional booster doses in this population.

## Figures and Tables

**Figure 1 viruses-16-01805-f001:**
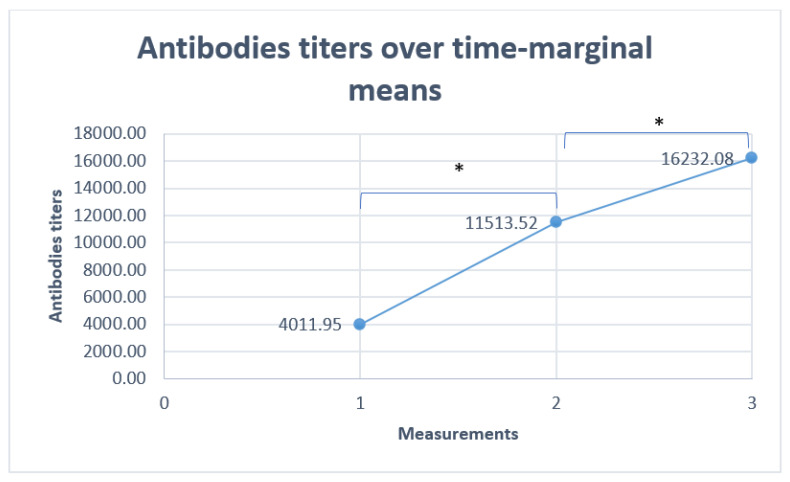
Antibody titers over time for 46 patients with three measurements. * *p* < 0.001.

**Table 1 viruses-16-01805-t001:** Demographic and clinical characteristics of the patients enrolled in our study.

Demographic and Clinical Characteristics	Total Number of PLVWH	PLVWH with CD4 < 200	PLVWH with 200 < CD4 < 500	PLVWH with CD4 ≥ 500	*p* Value
**Total number of PLWH (%)**	784 (100)	77 (9.8)	315 (40.2)	392 (50)	
**Females (%)**	348 (44.4)	27 (35.1)	127 (40.3)	194 (49.4)	*p* = 0.011
**Males (%)**	436 (55.6)	50 (64.9)	188 (59.7)	198 (50.5)	
**Mean age ± SD (range), years**	50.2 ± 12.2 (18–89)	53.2 ± 12.9 (22–89)	51.7 ± 13.1 (18–85)	48.4 ± 11.1 (20–89)	*p* = 0.0001
**Years since HIV diagnosis**	16.5 ± 7.8	17.4 ± 8	16.2 ± 7.7	16.6 ± 7.8	*p* = 0.388
**Nadir of CD4 cells/µL**	187.8 ± 170	79.4 ± 100.6	136.8 ± 98.1	249.9 ± 199.7	*p* = 0.001
**Viral load suppression (%)**	736 (93.9)	61 (79.2)	293 (93)	382 (97.4)	*p* = 0.0001

**Table 2 viruses-16-01805-t002:** COVID-19 infections, hospital admissions, and antibody titers.

Clinical and Laboratory Data		Total Number of PLVWH		PLVWH with CD4 < 200	PLVWH with 200 < CD4 <500	PLVWH with CD4 ≥ 500	*p* Value
**COVID-19 vaccination (at least two doses) (%)**		708 (90.3)		68 (88.3)	281(89.2)	359 (91.5)	
**COVID-19 infection (%)**		217 (27.6)		19 (24.6)	88 (27.9)	110 (28.1)	*p* = 0.824
**Patients hospitalized for COVID-19 infection (%)**		6 (0.8)		1 (1.3)	5 (1.68)	0 (0)	*p* = 0.047
**Patients with first antibody titer for** **COVID** **-19**	No immunity	784	166 (21.2)	32 (41.6)	62 (19.7)	72 (18.3)	*p* = 0.0001
	Normal titer		403 (51.4)	29 (37.6)	158 (50.2)	216 (55.1)	
	High titer		215 (27.4)	16 (20.7)	95 (30.2)	104 (26.5)	
**Patients with second antibody titers for** **COVID** **-19**	No immunity	555	62 (11.2)	11 (25.5)	25 (10.9)	26 (9.5)	*p* = 0.041
	Normal titer		159 (28.6)	18 (41.8)	71 (31)	70 (25.6)	
	High titer		334 (60.2)	24 (55.8)	133 (58.1)	177 (64.8)	
**Patients with third antibody titers for** **COVID** **-19**	No immunity	46	3 (6.5)	2 (28.6)	1 (5.2)	0 (0)	*p* = 0.038
	Normal titer		12 (26.1)	3 (42.8)	3 (15.7)	6 (30)	
	High titer		31 (67.4)	2 (28.6)	15 (78.9)	14 (70)	

**Table 3 viruses-16-01805-t003:** Multinomial logistic regression of age, sex, previous COVID-19 infection, and last CD4 value of first and second titer of COVID-19 antibodies.

	First Titer of COVID-19 Antibodies (784 Patients)					Second Titer of COVID-19 Antibodies (555 Patients)				
	B	Std. Error	Wald (df = 1)	Exp (B)	95% Confidence Interval	B	Std. Error	Wald	Exp (B)	N
**Age**	0.016	0.009	3.471	1.016	0.999–1.034	0.024	0.012	4.224 *	1.025	1.001–1.049
**Sex**	0.232	0.214	1.169	1.261	0.828–1.918	0.555	0.286	3.767	1.742	0.995–3.05
**COVID-19 infection**	0.214	0.244	0.769	1.238	0.768–1.996	−0.220	0.318	0.479	0.803	0.431–1.496
**CD4** **(<200)**	−1.189	0.349	11.609 *	0.304	0.154–0.603	−1.344	0.432	9.704 **	0.261	0.112–0.607

* *p* < 0.05, ** *p* < 0.01.

## Data Availability

The data presented in this study are available on request from the corresponding author due to privacy reasons.

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
