# Peer review of "COVID-19 Vaccinations, Infections, and Outcomes Among 784 People Living with HIV"

_viruses, 2024, doi:10.3390/v16121805_

Round 1

Reviewer 1 Report

Comments and Suggestions for Authors

This paper investigated the immunologic response to COVID-19 vaccination among PLWH. It evaluated COVID-19 morbidity, hospitalization and the magnitude of immunological response to sequential BNT 1626b2 mRNA vaccines in PLWH regarding demographic and clinical factors. The authors demonstrate that PLWH are responsive to COVID-19 vaccines, and patients with higher CD4 cells count have higher titers of COVID-19 antibodies. The findings underscore the need for continued efforts to vaccinate PLWH, particularly those with lower CD4 counts, to ensure broad protection against COVID-19.

There are some problems, which must be solved before it is considered for publication.

1. Even though this paper provides some clinical evidence, the conclusions obtained in this paper seem not novel. For example, the conclusion that “PLWH with lower CD4 levels have lower immune reactivity after being vaccinated with mRNA vaccine” has been reported in previous published studies.

2. Relevant research background needs to be supplemented in INTRODUCTION. There have been many studies on COVID-19 vaccination in PLWH and the clinical characteristics of their serologic response.

3. The phrase "less severe illness" has no basis in this paper in lines 27-28, because there is no data supporting the severity of the COVID-19 disease.

4. Although the method used for data analysis is appropriate, the description of statistical test results is still ambiguous. For example, in Table 3, concerning the multinomial logistic regression analysis, the CD4 cell count exhibited a negative correlation with the first titer of COVID-19 antibody (B=-1.189, odds ratio=0.261, 95% confidence interval=0.154-0.603), which is exactly the opposite of the conclusion presented in this paper. Please verify the statistical results.

5. The conclusion that “previous infection with COVID-19 did not have significant influence on first COVID-19 antibodies titers”contradicts the findings of Mary Duro et al. Please further discuss the reasons and consider whether this conclusion should be obtained by the longitudinal data before and after infection with COIVD-19.

6. Please explain whether the previous COVID-19 infection was breakthrough infection. Previous studies by Xueying Yang et al. found that higher CD4 count was significantly correlated with fewer breakthroughs in PLWH. Please discuss it carefully with related studies.

7. There are some spelling errors in the manuscript, such as, in Lines 15 and 138, “BNT 162b2 mRNA vaccines" would be “BNT 1626b2 mRNA vaccines". Additionally, the symbol placement in Figure 3 is inaccurate. Please check the manuscript carefully.

Author Response

Reviewer 1

  1. "Even though this paper provides some clinical evidence, the conclusions obtained in this paper seem not novel. For example, the conclusion that “PLWH with lower CD4 levels have lower immune reactivity after being vaccinated with mRNA vaccine” has been reported in previous published studies."

Our study is a real-life study from Israel that contribute to the growing body of evidence that suggests PLWH, particularly those with well-managed HIV, exhibit a robust immunologic response to COVID-19 vaccination.

  1. "Relevant research background needs to be supplemented in INTRODUCTION. There have been many studies on COVID-19 vaccination in PLWH and the clinical characteristics of their serologic response."

As was suggested we added details of studies regarding COVID-19 vaccination in PLWH and the clinical characteristics of their serologic response. (Page 4 , Paragraph 2,3; Page 5 ,Paragraph 1)

  1. "The phrase "less severe illness" has no basis in this paper in lines 27-28, because there is no data supporting the severity of the COVID-19 disease"

The phrase "less severe illness" was replaced by "lower hospitalization rate".

  1. "Although the method used for data analysis is appropriate, the description of statistical test results is still ambiguous. For example, in Table 3, concerning the multinomial logistic regression analysis, the CD4 cell count exhibited a negative correlation with the first titer of COVID-19 antibody (B=-1.189, odds ratio=0.261, 95% confidence interval=0.154-0.603), which is exactly the opposite of the conclusion presented in this paper. Please verify the statistical results"

In our multinomial logistic regression, we compared the group of patients with Low CD4 cell count in the last blood test to the group of patients with CD4 cell count between 200 to 500 and to the group of patients with more than 500 CD4 cell count and found that low CD4 has negative impact: reduces the chance to have normal titer by 73% and the chance to have high titer by 69.6%. to clarify this, we added the word low to Table 3.

  1. "The conclusion that “previous infection with COVID-19 did not have significant influence on first COVID-19 antibodies titers” contradicts the findings of Mary Duro et al. Please further discuss the reasons and consider whether this conclusion should be obtained by the longitudinal data before and after infection with COIVD-19."

As was suggested, we analyzed the time of COVID-19 infection in relation to the time of antibodies titers measurement and found that most of our COVID-19 infected patients, were infected after the first titer test. We added this information to the Results part (Page 8, Paragraph 3,4) of our article and discussed it further on the discussion part (Page 11; Paragraph 3) of our article. Thank you for this significant remark.

  1. "Please explain whether the previous COVID-19 infection was breakthrough infection. Previous studies by Xueying Yang et al. found that higher CD4 count was significantly correlated with fewer breakthroughs in PLWH. Please discuss it carefully with related studies."

The 217 patients with COVID-19 infection in our study included 81 patients who were infected before March 2021, 66 patients who were partially vaccinated and had COVID-19 infection during the study period and 70 patients who were fully vaccinated (3 doses) and were infected with COVID-19 during the study period these patients were defined as breakthrough infection. We added this information to the article (Page 9, Paragraph 1). We further discussed this in Page11, Paragraph 2

  1. "There are some spelling errors in the manuscript, such as, in Lines 15 and 138, “BNT 162b2 mRNA vaccines" would be “BNT 1626b2 mRNA vaccines". Additionally, the symbol placement in Figure 3 is inaccurate. Please check the manuscript carefully"

The spelling errors in the manuscript were corrected. (Lines 15, 51, 79, 129, 138).               The symbol placement in Figure 1 was corrected.

Reviewer 2 Report

Comments and Suggestions for Authors

This paper describes the humoral immunity to SARS-CoV-2 of over 700 Israelis living with HIV who had received the same vaccine against COVID-19. At least one serological test was performed during a limited period of 2021. 

This large study is interesting since previous reports on this population  did not generally include so many HIV infected patients. 

However, important data is missing, concerning the delay between vaccine doses and the time when serological tests were carried out. 

These data could be used to analyse:

- why Ab titers increased over time, when it is known that antibody titers increase during the first month following a vaccine dose, but then tend to decrease 

- why 20 to 30% of the patients have high Ab titers on the first test : is this related to previous covid infection, given that the number in each group for "high titer" and "covid infection" are closed, or is there a relationship with those having received 3 vaccine doses ?

Given the real-life nature of this study the population is likely to be heterogeneous, justifying the need for such data to better interpret results by CD4 cell number or range.

It is not described how covid infection was defined and when it occurred prior to testing. As the authors pointed out, their antibody test does not distinguish between vaccine-induced and infection-induced antibodies, and it is well known that antibody response to the vaccine is higher in previously infected people than in people who have never been infected. This can alter the distribution of antibody titers independently of which group they belong to. It is therefore necessary to indicate the proportion of previously infected individuals in each antibody level group.

With regard to the classification of antibody titers into 3 groups, in the method section it is noted that titers below 50 are considered to indicate no immunity and between 50 and 40000 are considered to reflect an immunity, while in table 2  there are a group of low titer and a group of normal titer. Did the authors consider these 2 descriptions to be equivalent ? if so, it would be wrong to consider the absence of immunity as a low titer. If this is not the case, the authors should give their definition in the legend to table 2 .

Finally, in table 1, the total percentage of women and men does not add up to 100%.

Author Response

Reviewer 2

  1. "This paper describes the humoral immunity to SARS-CoV-2 of over 700 Israelis living with HIV who had received the same vaccine against COVID-19. At least one serological test was performed during a limited period of 2021. This large study is interesting since previous reports on this population did not generally include so many HIV infected patients. However, important data is missing, concerning the delay between vaccine doses and the time when serological tests were carried out. These data could be used to analyse:

- why Ab titers increased over time, when it is known that antibody titers increase during the first month following a vaccine dose, but then tend to decrease 

 The increase in the titer of antibodies is probably due to repeated doses of the BNT 162b2 mRNA vaccine overtime, as only 4 patients in the group of 46 patients with three measurements of antibodies titer were diagnosed as positive for COVID-19. The time between measurements was short and close to vaccines (3 tests in 8 months, average of 35.6±24.1 days from third vaccine). However, undiagnosed asymptomatic COVID-19 infection in parallel to the spread of the COVID-19 pandemic cannot be ruled out. We clarified this in the text  (Page 11, last Paragraph).

why 20 to 30% of the patients have high Ab titers on the first test : is this related to previous covid infection, given that the number in each group for "high titer" and "covid infection" are closed, or is there a relationship with those having received 3 vaccine doses ?

High titer of antibodies in 27% of the patients is probably related to repeated doses of vaccination against COVID-19, since the rate of previous to first measurement COVID-19 infection was low in this group (10/215, 4.6%). (Page 8 , Paragrah 4)

Given the real-life nature of this study the population is likely to be heterogeneous, justifying the need for such data to better interpret results by CD4 cell number or range."

  1. It is not described how covid infection was defined and when it occurred prior to testing. As the authors pointed out, their antibody test does not distinguish between vaccine-induced and infection-induced antibodies, and it is well known that antibody response to the vaccine is higher in previously infected people than in people who have never been infected. This can alter the distribution of antibody titers independently of which group they belong to. It is therefore necessary to indicate the proportion of previously infected individuals in each antibody level group.

COVID-19 infection was defined as positive result of RT-PCR testing in nasopharyngeal swab. The tests were performed by ministry of health approved laboratories in Israel and the positive results appeared on the electronic medical file of the patients. As was suggested, we added this information to the methods part of our article (Page 6, Paragraph 2).

As was suggested, we analyzed the time of COVID-19 infection in relation to the time of antibodies titers measurement and found that most of our COVID-19 infected patients, were infected after the first titer test. We added this information to the Results part (Page 8, Paragraph 3,4) of our article.  

As was suggested, we analyzed the proportion of previously infected COVID-19 patients in each antibody level group. We discussed it further on the discussion part (Page 11; Paragraph 3) of our article. Thank you for this significant remark.

  1. With regard to the classification of antibody titers into 3 groups, in the method section it is noted that titers below 50 are considered to indicate no immunity and between 50 and 40000 are considered to reflect an immunity, while in table 2  there are a group of low titer and a group of normal titer. Did the authors consider these 2 descriptions to be equivalent ? if so, it would be wrong to consider the absence of immunity as a low titer. If this is not the case, the authors should give their definition in the legend to table 2 .

According to your suggestion low titer was replaced with "no immunity" (Table 2).

  1. Finally, in table 1, the total percentage of women and men does not add up to 100%.

The mistake in Table 1 line 3 was corrected.  

Round 2

Reviewer 2 Report

Comments and Suggestions for Authors

The authors have answered my questions. Only one thing remains to be corrected: the term "no immunization" should be replaced by "no immunity" as vaccination is immunization.

Author Response

Corrected. Thanks.